# Evaluation of Proteasome and Immunoproteasome Levels in Plasma and Peritoneal Fluid in Patients with Endometriosis

**DOI:** 10.3390/ijms241814363

**Published:** 2023-09-21

**Authors:** Monika Wróbel, Zielińska Zuzanna, Łukasz Ołdak, Aleksandra Kalicka, Grzegorz Mańka, Mariusz Kiecka, Robert Z. Spaczyński, Piotr Piekarski, Beata Banaszewska, Artur Jakimiuk, Tadeusz Issat, Wojciech Rokita, Jakub Młodawski, Maria Szubert, Piotr Sieroszewski, Grzegorz Raba, Kamil Szczupak, Tomasz Kluz, Marek Kluza, Piotr Pierzyński, Cezary Wojtyła, Michał Lipa, Damian Warzecha, Mirosław Wielgoś, Włodzimierz Sawicki, Ewa Gorodkiewicz, Piotr Laudański

**Affiliations:** 11st Department of Obstetrics and Gynecology, Medical University of Warsaw, 02-015 Warsaw, Poland; 2Bioanalysis Laboratory, Doctoral School of Exact and Natural Science, Faculty of Chemistry, University of Bialystok, Ciolkowskiego 1K, 15-245 Bialystok, Poland; z.zielinska@uwb.edu.pl (Z.Z.); l.oldak@uwb.edu.pl (Ł.O.); 3Bioanalysis Laboratory, Faculty of Chemistry, University of Bialystok, Ciolkowskiego 1K, 15-245 Bialystok, Poland; ewka@uwb.edu.pl; 4Faculty of Medicine, Medical University of Warsaw, Żwirki i Wigury 61, 02-091 Warsaw, Poland; ak.lexy.98@gmail.com; 5Angelius Provita Hospital, 40-611 Katowice, Poland; grzegorzmanka@orange.pl (G.M.); m.kiecka@angelius.org (M.K.); 6Center for Gynecology, Obstetrics and Infertility Treatment Pastelova, 60-198 Poznan, Poland; rspaczynski@yahoo.com; 7Division of Infertility and Reproductive Endocrinology, Department of Gynecology, Obstetrics and Gynecological Oncology, Poznan University of Medical Sciences, 61-701 Poznan, Poland; piotr-p90@wp.pl; 8Chair and Department of Laboratory Diagnostics, Poznan University of Medical Sciences, 61-701 Poznan, Poland; bbeata48@gmail.com; 9Department of Reproductive Health, Institute of Mother and Child in Warsaw, 01-211 Warsaw, Poland; jakimiuk@yahoo.com; 10Department of Obstetrics and Gynecology, Central Clinical Hospital of the Ministry of Interior, 02-507 Warsaw, Poland; 11Department of Obstetrics and Gynecology, Institute of Mother and Child in Warsaw, 01-211 Warsaw, Poland; tadeusz.issat@imid.med.pl; 12Collegium Medicum, Jan Kochanowski University in Kielce, 25-516 Kielce, Polandkuba.mlodawski@gmail.com (J.M.); 13Clinic of Obstetrics and Gynecology, Provincial Combined Hospital in Kielce, 25-736 Kielce, Poland; 14Department of Gynecology and Obstetrics, Medical University of Lodz, 90-419 Lodz, Poland; maria.szubert@gmail.com (M.S.); sieroszewski@gmail.com (P.S.); 15Department of Surgical Gynecology and Oncology, Medical University of Lodz, 90-419 Lodz, Poland; 16Department of Fetal Medicine and Gynecology, Medical University of Lodz, 90-419 Lodz, Poland; 17Clinic of Obstetric and Gynecology in Przemysl, 37-700 Przemysl, Poland; rabagrzegorz@gmail.com (G.R.); fisk1986@gmail.com (K.S.); 18Department of Obstetrics and Gynecology, University of Rzeszow, 35-310 Rzeszow, Poland; 19Department of Gynecology, Gynecology Oncology and Obstetrics, Institute of Medical Sciences, Medical College of Rzeszow University, 35-310 Rzeszow, Poland; jtkluz@interia.pl (T.K.); marek.kluza@gmail.com (M.K.); 20OVIklinika Infertility Center, 31 Połczyńska Street, 01-377 Warsaw, Poland; piotr.pierzynski@gmail.com (P.P.); damwarzecha@gmail.com (D.W.); 21City South Hospital Warsaw, 02-781 Warsaw, Poland; michallipa1@gmail.com; 22Premium Medical Clinic, 04-359 Warsaw, Poland; miroslaw.wielgos@gmail.com; 23Medical Faculty, Lazarski University, 02-662 Warsaw, Poland; 24Department of Obstetrics, Gynecology and Gynecological Oncology, Medical University of Warsaw, 03-242 Warsaw, Poland; saw55@wp.pl; 25Women’s Health Research Institute, Calisia University, 62-800 Kalisz, Poland

**Keywords:** endometriosis, proteasome, immunoproteasome, plasma, peritoneal fluid

## Abstract

Endometriosis is a chronic disease in which the endometrium cells are located outside the uterine cavity. The aim of this study was to evaluate circulating 20S proteasome and 20S immunoproteasome levels in plasma and peritoneal fluid in women with and without endometriosis in order to assess their usefulness as biomarkers of disease. Concentrations were measured using surface plasmon resonance imaging biosensors. Patients with suspected endometriosis were included in the study—plasma was collected in 112 cases and peritoneal fluid in 75. Based on the presence of endometriosis lesions detected during laparoscopy, patients were divided into a study group (confirmed endometriosis) and a control group (patients without endometriosis). Proteasome and immunoproteasome levels in both the plasma (*p* = 0.174; *p* = 0.696, respectively) and the peritoneal fluid (*p* = 0.909; *p* = 0.284, respectively) did not differ between those groups. There was a statistically significant difference in the plasma proteasome levels between patients in the control group and those with mild (Stage I and II) endometriosis (*p* = 0.047) and in the plasma immunoproteasome levels in patients with ovarian cysts compared to those without (*p* = 0.017). The results of our study do not support the relevance of proteasome and immunoproteasome determination as biomarkers of the disease but suggest a potentially active role in the pathogenesis of endometriosis.

## 1. Introduction

Endometriosis is an enigmatic, estrogen-dependent disease in which endometrial tissue, including glands and stroma, is located outside the uterus. This chronic inflammatory condition is classified as benign; however, it involves malignant behavior of invasion and migration [1,2]. Endometriosis is estimated to affect 10% of women of reproductive age [3]. It manifests in a variety of nonspecific symptoms, including gynecological (dysmenorrhea, pelvic pain, dyspareunia), digestive (diarrhea and/or constipation, pain on bowel movement, intestinal cramping, pain on defecation, cyclic rectal bleeding), lower back pain and asthenia [4]. Diagnosis can be challenging and a large diagnostic still exists, delay reaching up to 7.4 years [3].

One the most well-known theories about the mechanism underlying endometriosis is retrograde menstruation, proposed by Sampsons. It states that menstrual blood containing endometrial cells flows back through the fallopian tubes. Migrated endometrial cells cause an inflammatory reaction, fibrosis and pain [5]. This theory has not been proven and the pathogenesis of the disease is still being investigated, including the involvement of molecular factors such as proteins, proteases and autoantibodies [6,7,8,9].

Laparoscopy is no longer the diagnostic gold standard and it is only recommended in patients with negative imaging results and/or where empirical treatment was unsuccessful [10]. Currently, clinical and imaging examination are used depending on the suspicion of endometriosis. Clinicians are recommended to use imaging techniques (ultrasound or magnetic resonance imaging) in the diagnostic scheme, but a negative finding does not exclude endometriosis [10]. There is therefore a great need to search for new diagnostic methods, among which biomarkers may have potential value.

The primary function of proteasomes and immunoproteasomes is to remove proteins that are malformed, damaged by stress conditions or that need to be degraded by standard turnover [11]. Proteasomes and immunoproteasomes are involved in the intracellular proteolysis of proteins, including those associated with cell cycle control and apoptosis regulation, proteins encoded by tumor suppressor genes and those involved in the immune response. This intracellular proteolysis is not random, affecting proteins marked by ubiquitin attachment [12].

The immunoproteasome is formed after proteasome activation by proinflammatory cytokines, such as interferon (IFN)-γ and tumor necrosis factor (TNF) [13,14]. The active form of the proteasome and the immunoproteasome is the 26S complex, which contains two subcomplexes: the regulatory 19S responsible for substrate recognition and translocation to the 20S subunit, which forms the core of the molecule [12]. Both the constitutive 20S proteasome and the 20S immunoproteasome are composed of four rings, each containing seven distinct subunits. The outer α rings control protein entry into the central catalytic compartment, while the internal β rings contain the subunits responsible for peptidase activities: caspase-like, trypsin-like and chymotrypsin-like [15]. In the immunoproteasome, substitution of selected β subunits for cytokine-induced homologues results in a reduction in caspase activity, while trypsin and chymotrypsin activity is enhanced [16]. As a result, the immunoproteasome degrades proteins, producing peptides suitable for major histocompatibility complex class I (MHC-I) presentation [17]. Consequently, immunoproteasomes are involved in T cell expansion and T helper cell differentiation, leading to their role in the pathogenesis of inflammatory diseases [11].

The intracellular immunoproteasome is expressed in autoimmune diseases, hematological malignancies and cancer, suggesting that its role is not limited to antigen presentation [17,18]. Due to the potential role of the immunoproteasome in the pathogenesis of these diseases, the use of immunoproteasome inhibitors is currently being investigated for their treatment [19]. Intracellular proteasome inhibitors such as bortezomib, carfilzomib and ixazomib are successfully used in the treatment of multiple myeloma [20].

Recently, it has been demonstrated that 20S proteasomes are located not only intracellularly, but also in the extracellular space, for example in plasma. Circulating 20S proteasomes are physiologically present in human plasma, while elevated levels have been found in blood cancers, solid tumors, autoimmune diseases, trauma and sepsis [21]. The origin of extracellular proteasomes is unknown. No transporter has been found to facilitate the passage of such large complexes across the cell membrane. Extracellular proteasomes may originate from blood cells, endothelial cell cytolysis or be released into plasma as a result of the breakdown of extracellular vesicles [22].

Endometriosis develops under conditions of inflammation; recent studies have also indicated its association with autoimmune diseases [23]. In the presence of endometriosis, the risk of certain malignancies, such as ovarian and thyroid cancer, is increased [24]. The potential role of the proteasome and immunoproteasome in the pathogenesis of autoimmune diseases and cancer and the association of these diseases with endometriosis suggests that these protein complexes may also be involved in the development of endometriosis.

Surface plasma resonance imaging (SPRi) is a novel spectroscopic technique used to detect local refractive index changes after molecules bind to the surface. This surface is made of glass coated with a thin layer of metal (e.g., gold) and a receptor, which forms a layer of active biomolecules. In contrast to the commonly used Elisa method, SPRi is direct and does not require the label that can change the properties of the investigated protein [15,25,26].

The aim of this study was to compare the levels of circulating proteasome and immunoproteasome in plasma and peritoneal fluid using SPRi in women with endometriosis and in a control group in order to assess their role in the pathogenesis of the disease and to evaluate them as its potential biomarkers.

## 2. Results

Table 1 and Table 2 show the characteristics of the patients in whom plasma and peritoneal fluid were collected, respectively. Both groups were homogeneous and adequately matched. The only significant statistical difference between the groups was the presence of an ovarian cyst (*p* < 0.001). There were no statistically significant differences in the proteasome and immunoproteasome concentrations in any of the studied body fluids. The distribution of concentrations of the proteasome and immunoproteasome in plasma and peritoneal fluid are presented in Figure 1 and Figure 2, respectively.

Table 3 shows the plasma proteasome and immunoproteasome concentrations in the different stages of endometriosis and in the control group. Table 4 shows the comparison of these concentrations in the groups with mild (Stage I and II) and severe endometriosis (Stage III and IV) and in the control group. The results are graphically presented in Figure 3. There was a statistically significant difference between proteasome levels in Stage I and Stage II endometriosis compared to the control group (*p* = 0.047). There was a clear tendency for plasma proteasome and immunoproteasome levels to increase in more advanced stages of endometriosis, but the difference was not statistically significant (*p* = 0.095 and *p* = 0.062, respectively). 

In Table 5 and Table 6, analogous determinations were made in the peritoneal fluid. We found no statistically significant differences between proteasome and immunoproteasome concentrations according to the presence and severity of endometriosis. A graphical representation of these results is shown in Figure 4.

Finally, we compared the concentrations of the studied parameters in relation to the presence of an ovarian cyst, history of infertility and cycle phase. The results we obtained are presented in Table 7 and Table 8. Plasma immunoproteasome concentrations in patients with an ovarian cyst were statistically significantly higher than in patients in whom ovarian cyst was not present (*p* = 0.017). It is worth noting that although all patients diagnosed with an ovarian cyst were also diagnosed with endometriosis, not all of these cysts turned out to be endometrial cysts after histopathological evaluation.

## 3. Discussion

To our knowledge, this is the first study to simultaneously evaluate proteasome and immunoproteasome concentrations in the plasma and peritoneal fluid of patients with endometriosis. Our results showed no differences in the concentrations of these proteins in the studied body fluids between women with endometriosis and the control group, thus negating their use as biomarkers of this disease.

However, our findings do not exclude the potential role of the proteasome and immunoproteasome in the pathogenesis of endometriosis. The main result we obtained is a statistically significant reduced plasma proteasome concentration between patients with mild endometriosis (Stage I and Stage II) and the control group. This could be explained by the different effects of oxidative stress on proteasome activity depending on its intensity. Mild oxidative stress, present in the early stages of endometriosis [27], stimulates proteasome activity and the ubiquitination pathway, resulting in the accumulation of proteasome and ubiquitin conjugates in the cell [28]. Although the mode of proteasome transport from the cell to the extracellular space is not known [21], studies in recent years show that there are mechanisms of trans-membrane translocation even for large complexes [29]. Based on these findings, it can be hypothesized that the proteasome can be actively transported across the cell membrane, but its intracellular conjugation with ubiquitin inhibits translocation to the plasma due to the excessive size of the complex, resulting in a decrease in the circulating free proteasome. The severe stress present in advanced stages of the disease inhibits proteasome activity so that ubiquitin complexes are not formed and plasma proteasome levels remain constant.

Although present in plasma, the statistically significant reduction in proteasome levels in mild endometriosis compared to controls was not observed in peritoneal fluid. Oxidative stress in peritoneal fluid is initiated in inflammatory cells and the products of this process are transported to plasma. Consequently, peritoneal fluid is more sensitive to the effects of oxidative stress than plasma [27], thus missing the effect of mild oxidative stress in the early stages of endometriosis.

In contrast to the proteasome, we did not observe a reduction in plasma immunoproteasome levels in mild endometriosis. The onset of endometriosis can be divided into two stages: an initial immune-dependent stage and a later stage in which estrogen signaling dominates [30]. More inflammatory markers are secreted in the early stages of endometriosis [31], including cytokines that stimulate immunoproteasome formation. Exposure to pro-inflammatory cytokines induces protein oxidation, which then undergoes polyubiquitination. This results in a temporary accumulation of polyubiquitin conjugates inside the cell [32] and their subsequent degradation by immunoproteasomes [32,33]. Transport across the cell membrane of such large complexes as immunoproteasome and polyubiquitin conjugates can be impaired. Despite this, the continuous production of the immunoproteasome in the early stages of endometriosis is presumed to maintain its constant levels. 

Endometriosis has much in common with the neoplastic process and proteasome inhibitors have proven effective for both these diseases. As an example, the proteasome inhibitor bortezomib is successfully used as anti-cancer drug. A similar effect has been achieved in an animal model of endometriosis—the proteasome inhibitor disulfiram was proven to reduce the size of artificially induced endometrial lesions [34]. Indeed, there is much more evidence of the similarity between endometriosis and the cancer process. Somatic mutations of cancer-associated genes are present in deep-infiltrating endometriosis and endometriotic cysts [35]. A study by Anglesio et al. found that more than a quarter of lesions in deeply infiltrating endometriosis contained cancer driver mutations [36]. There is an association between endometriosis and ovarian cancer, particularly with its two subcategories: endometrioid and clear-cell carcinoma [37,38].

Henry L et al., using cirrhosis as an example, showed that elevated plasma proteasome levels are associated with neoplastic transformation [39]. Therefore, based on the similarity between endometriosis and the cancer process, we expected elevated proteasome levels in more advanced stages of endometriosis, but our results showed no statistically significant differences. The reduction in plasma proteasome levels in mild endometriosis compared to the control group may instead support the theory that the early stages of the disease are a manifestation of oxidative stress [40]. It is also worth noting that proteasome inhibitors act on the intracellular pathway, and the concentration of their extracellular form may be influenced by the mechanism of transmembrane transport.

In addition, we have demonstrated that in the presence of an ovarian cyst, the plasma concentration of the immunoproteasome is statistically significantly higher. This relationship was not shown for the proteasome, thus a role for the inflammatory component in cyst formation might be expected. The inflammatory process affects follicular dynamics and ovulation [41]. Increased plasma concentrations of inflammatory markers, including TNF-α and Interleukin 6 (IL-6), have been shown, for example, in polycystic ovary syndrome [42]. A possible role for the inflammatory process present in plasma in the formation of ovarian cysts in patients with endometriosis is therefore possible. This relationship has not been demonstrated in peritoneal fluid, which may be explained by the rich vascularization of the ovary and the greater influence of inflammatory markers in plasma on its function.

We have also analyzed proteasome and immunoproteasome concentrations according to cycle phase and the presence of infertility factor in patients with and without endometriosis. In both cases, we did not observe differences, either in plasma or peritoneal fluid.

The role of oxidative stress in menstrual cycle phases has not yet been determined. The literature reports that it increases around the estrogen peak [43] and, depending on the reference source, increases [44] or decreases [43] during the luteal phase. Recent reports also show that the inflammatory factor has an influence on the length of the menstrual cycle [45]. Due to the complex hormonal relationship and the potential role of oxidative stress and the inflammatory factor on the menstrual cycle, the validity of proteasome and immunoproteasome determinations in its course, both in patients with and without endometriosis, requires further research. Similarly, in the case of infertility, its multifactorial etiology, especially in women with endometriosis [46], and the potential role of an inflammatory factor [47] do not exclude the validity of proteasome and immunoproteasome determinations. 

We are aware that the relatively small number of samples is a weakness of this study. Further studies based on a multicenter patient base in Poland are already planned, in which we expect to extend the analysis. The strength of our study is its innovative nature. By assessing the concentrations of both the proteasome and the immunoproteasome in plasma and peritoneal fluid, we were able to evaluate their inter-relationships and potential impact on the pathogenesis of endometriosis. Another advantage of our research was the very careful sampling procedure, with particular attention being paid to the purity of the peritoneal fluid collected. It is also worth noting that our study is part of a series of publications evaluating the use of selected molecules as biomarkers of endometriosis.

## 4. Materials and Methods

### 4.1. Study Population

This study is part of a multicenter project conducted in 8 centers in Poland between 2018 and 2019: Department of Obstetrics and Gynecology, Medical University of Warsaw; Angelius Provita Hospital in Katowice; Department of Gynecology, Division of Infertility and Reproductive Endocrinology, Obstetrics and Gynecological Oncology at Poznan University of Medical Sciences; Department of Obstetrics and Gynecology, Central Clinical Hospital of the Ministry of Interior in Warsaw; Clinic of Obstetrics and Gynecology, Provincial Combined Hospital in Kielce; Department of Surgical Gynecology and Oncology, Medical University of Lodz; Department of Gynecology and Obstetrics, Provincial Hospital in Przemysl; Department of Gynecology, Gynecology Oncology, and Obstetrics, Institute of Medical Sciences, Medical College of Rzeszow University. The study was approved by the Ethics Committee of the Medical University of Warsaw (KB/223/2017).

Our cohort comprised women between 18 and 40 years qualified for planned laparoscopic surgery due to at least one condition: infertility, chronic pelvic pain or ovarian cysts. The exclusion criteria included malignant disease, previous and/or current pelvic inflammatory disease, irregular menstruation (less than 25 days or more than 35), hormone therapy within three months preceding laparoscopy, history of pelvic surgery, polycystic ovaries and uterine fibroids. A detailed description of the patient recruitment process is described in our most recent article [6]. Before the operation, all the patients completed the World Endometriosis Research Foundation (WERF) clinical questionnaire and provided written informed consent to participate in the study. 

All women entered into the study underwent laparoscopy. Endometrial lesions found during surgery were assessed by the WERF EPHect Minimal Surgery Form and then examined histopathologically. Patients with confirmed endometriosis were divided into stages (I-IV) according to the American Society of Reproductive Medicine classification. Subsequently, due to the similarities in disease activity between stages, patients in stages I and II were included in one group and defined as a mild form of the disease. Similarly, patients in Stage III and Stage IV were placed in one group and defined as having a severe form of endometriosis. The control group consisted of patients who were not diagnosed with endometriosis during the laparoscopy. A flowchart summarizing the patient selection process has been recently published [6].

All patients enrolled to the study underwent gynecological examination and vaginal ultrasound before referral to surgery. The phase of the cycle was determined by the date of the last menstrual period and its average duration. In addition, it was confirmed via histological evaluation of the eutopic endometrial samples collected simultaneously through laparoscopy. The peripheral blood samples were collected before the surgery, prior to anesthesia, stored in ethylenediaminetetraacetic acid (EDTA) 10 mL tubes (Sarstedt), centrifuged at 2500× *g* for 10 min at 4 °C and divided into 500 mL tubes. The peritoneal fluid collection process was very precise in order to eliminate the possibility of blood contamination. The material was collected immediately after the start of the operation by experienced gynecologists using a Veress needle and then centrifuged at 1000× *g* for 10 min at 4 °C. Finally, the supernatant was transferred to a fresh 10 mL tube (Sarstedt) and also divided into 500 mL tubes. Both plasma and peritoneal fluid were then stored at −80 °C. It is worth noting that, in each case, the time between collection of body fluids did not exceed 45 min. The material collection had no influence on the medical management of the patients and was carried out in accordance with the Declaration of Helsinki. All samples were transported on dry ice to the Department of Obstetrics and Gynecology in Warsaw and then to the University of Bialystok, where the necessary measurements were performed.

### 4.2. Reagents

The following reagents were used for the tests: proteasome 20S enzyme complex (AFFINITI Research Products Ltd., Exeter, UK) and 20Si immunoproteasome (BIOMOL, Hamburg, Germany) as a standard solutions for calibration, PSI proteasome inhibitor (Z-Ile-Glu(OBut)-Ala-Leu-H) (BIOMOL, Hamburg, Germany), ONX-0914 immunoproteasome inhibitor (SelleckChem, Houston, TX, USA), EDC (N-ethyl-N′-(3-dimethylaminopropyl) carbodiimide) (Sigma, Steinheim, Germany), NHS (N-hydroxysuccinimide) (Aldrich, Munich, Germany), carbonate buffer pH = 8.5, 2-aminoethanethiol (cysteamine) (Aldrich, Munich, Germany), 1-octadecanethiol (ODM) (Aldrich, Munich, Germany), human albumin (SIGMA, Steinheim, Germany), absolute ethyl alcohol 99.8% (POCh, Gliwice, Poland), HBS-ES solution (pH = 7.40, 0.01 M HEPES, 0.15 M sodium chloride, 0.005% Tween-20, 3 mM EDTA) and PBS (pH = 7.40, phosphate-buffered saline) (BIOMED, Tokyo, Japan). The base of the biosensor is a plate with a gold layer (Ssens, Enschede, The Netherlands).

### 4.3. Measurement Steps

#### 4.3.1. Structure of the Biosensor Base

A detailed description of the structure of the biosensor base used is described in the article published by Gorodkiewicz et al. [25].

#### 4.3.2. Preparation of the Chip for the Determination of the Proteasome 20S in the Sample

For the 20S-PSI proteasome inhibitor, covalent immobilization was used. A layer of linker—cysteamine—was immobilized on the prepared biosensor by immersing the chip in an alcoholic 20 mM solution of cysteamine for 12 h. After washing with water and ethyl alcohol, the chip was dried in a stream of argon. The next step was the activation of the ligand that would capture the proteasome from the sample solutions—addition of EDC (15.6 µL), NHS (15.6 µL) and carbonate buffer (6.25 µL) to 3.25 µL of PSI inhibitor (an 80 nM PSI solution was used). Then, the activated receptor (PSI) was placed on a thiol (cysteamine)-modified surface and incubated at 37 °C for 1 h. To remove unbound biomolecules, the biosensor surface (active sites) was washed with HBS-ES buffer and distilled water. In order to eliminate non-specific adsorption, after incubation of the chip with the receptor, a BSA solution with a concentration of 1 ng/mL was applied to the active sites of the biosensor, and then washed several times with distilled water.

#### 4.3.3. Preparation of the Chip for the Determination of the 20Si Immunoproteasome in the Sample

Hydrophobic-type immobilization was used for the ONX0914 inhibitor. The first step was formation of a 1-octadecanothiol (ODM) monolayer via immersion of the chip in a 20 mM alcoholic solution of ODM for at least 12 h. Next, the chip was washed with distilled water, ethyl alcohol and dried in a stream of argon. The second step was the binding of ONX0914 with the ODM via formation of hydrophobic interactions. The inhibitor (ONX0914) in PBS buffer at a concentration of 15 µg/mL was placed on the thiol (ODM) surface and incubated at 37 °C for 24 h. The surface was then rinsed with distilled water and HBS-ES buffer, then a 1 ng/mL BSA solution was applied and washed again with water.

#### 4.3.4. SPRi Measurements

SPRi measurements were performed on the apparatus for surface plasmon resonance imaging (SPRi), available at the Bioanalysis Laboratory, Faculty of Chemistry, University of Bialystok. 

The SPRi apparatus is composed of a system of polarizers and lenses, as well as a diode laser that emits light with a wavelength of λ = 635 nm. Another feature of the SPRi device is a prism with a biosensor and a CCD camera gathering the reflected light, which is converted into an image. Surface plasmon resonance in the image version examines the change in the intensity of the reflected monochromatic and p-polarized light after applying successive layers that make up the biosensor. Due to the proportionality of the SPRi signal, i.e., the light intensity to the immobilized mass, it was possible to quantitatively test the analyte content in the sample after performing the appropriate mathematical operations. 

Biosensor preparation was described in the preceding paragraphs. The prepared chip with the ligand layer was placed on the prism of the SPRi device with oil immersion and the appropriate angle was selected. The structural elements of the SPRi device are movable (with an angular range of 24° to 75°), and the value of the aforementioned angle was selected separately for each biosensor used. The SPRi ligand signal was measured. Then, 3 µL of samples containing proteasome/immunoproteasome (depending on the used biosensor with the appropriate receptor layer) were applied and left for 10 min. After washing with distilled water and HBS-ES buffer (in order to remove unbound molecules from the surface), further measurements were taken and saved as images. 

The SPRi images were evaluated and the numerical signals were transformed into an analytically useful quantitative signal, i.e., the concentration of the proteasome and immunoproteasome tested with ImageJ software (version 1.53, National Institutes of Health, NIH). Concentrations for the 20S proteasome and 20Si immunoproteasome were determined on the basis of calibration curves determined immediately before the measurements, considering the appropriate dilutions. The calibration curves for the 20Si immunoproteasome and the sum of the 20S proteasome and the 20Si immunoproteasome, which are shown in Figure 5 and Figure 6, were used for the calculations. The samples were diluted twice with PBS buffer to fall within the linear concentration range. The linear range for the immunoproteasome covers concentrations from 0.35 to 4.2 µg/mL. The 20S proteasome concentration is equal to the concentration difference between the sum of the 20S proteasome and the 20Si immunoproteasome determined via the same method using a calibration curve. The linear range of the sensor is 0.5 to 6 µg/mL. 

The SPRi signal, which was proportional to the mass of entrapped proteasome or immunoproteasome, was obtained as the difference between the signals before and after interaction with the analyzed sample for each spot separately.

As mentioned before, each of the test samples was analyzed two times, and the obtained concentration value was the average of the obtained measurements.

### 4.4. Statistical Analysis

Statistical analysis was performed using STATISTICA 13.0 (TIBCO Software Inc.). Qualitative variables are presented in the form of tables with counts and their percentage within groups. Associations between characteristics were analyzed using the χ2P and χ2Y test (Yates correction for low counts in subgroups). Differences in the percentage of women with infertility in the groups were analyzed with a test for two structure indicators. For quantitative variables, basic measures of position and variability were calculated. Conformity to a normal distribution was tested using the Shapiro–Wilk test. For variables with a normal distribution, Student’s *t* test was used, and for variables that did not meet assumptions about the normality of the distribution, the Mann–Whitney U test was used. Selected group comparisons (pooled groups 1–2 vs. 3–4 degree of endometriosis; 1–2 degree of endometriosis vs. control; and 3–4 degree of endometriosis vs. control) were performed using contrast analysis. Statistical significance level was defined as *p* < 0.05.

## 5. Conclusions

Despite promising results regarding the association of the proteasome and immunoproteasome with the occurrence of many diseases, the relevance of the concentration of these proteins as a biomarker of endometriosis has not been proven in our study. Nevertheless, our results suggest an important role for these proteins in the pathogenesis of endometriosis, which definitely needs to be confirmed in further studies.

## Figures and Tables

**Figure 1 ijms-24-14363-f001:**
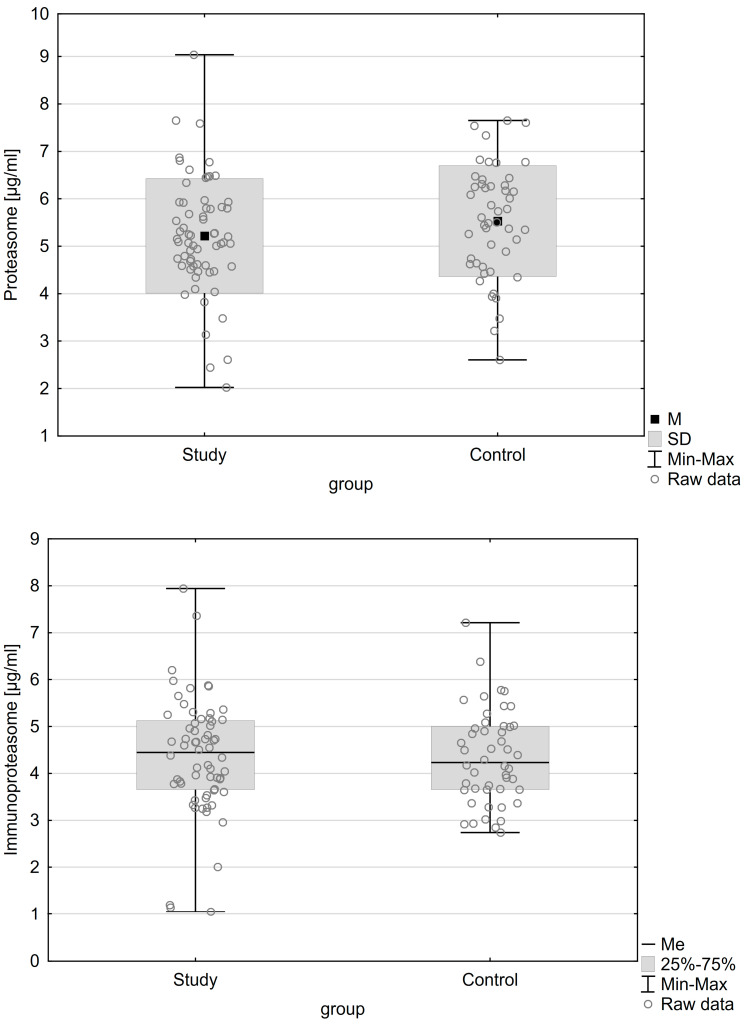
Distribution of concentrations of the studied parameters in plasma.

**Figure 2 ijms-24-14363-f002:**
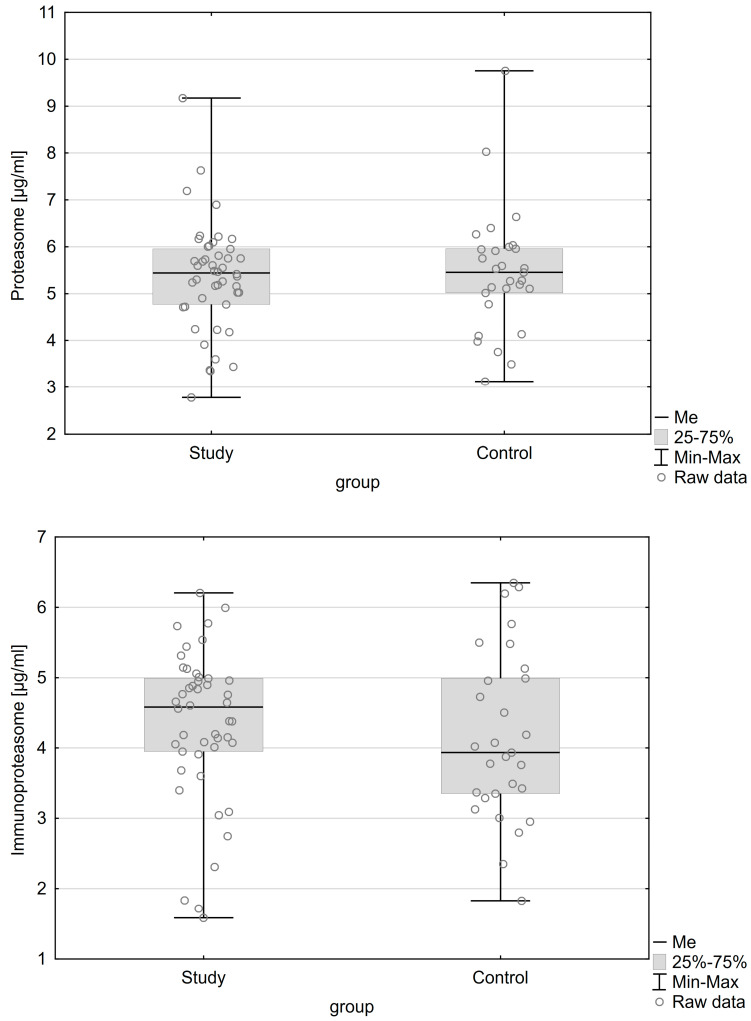
Distribution of concentrations of the studied parameters in peritoneal fluid.

**Figure 3 ijms-24-14363-f003:**
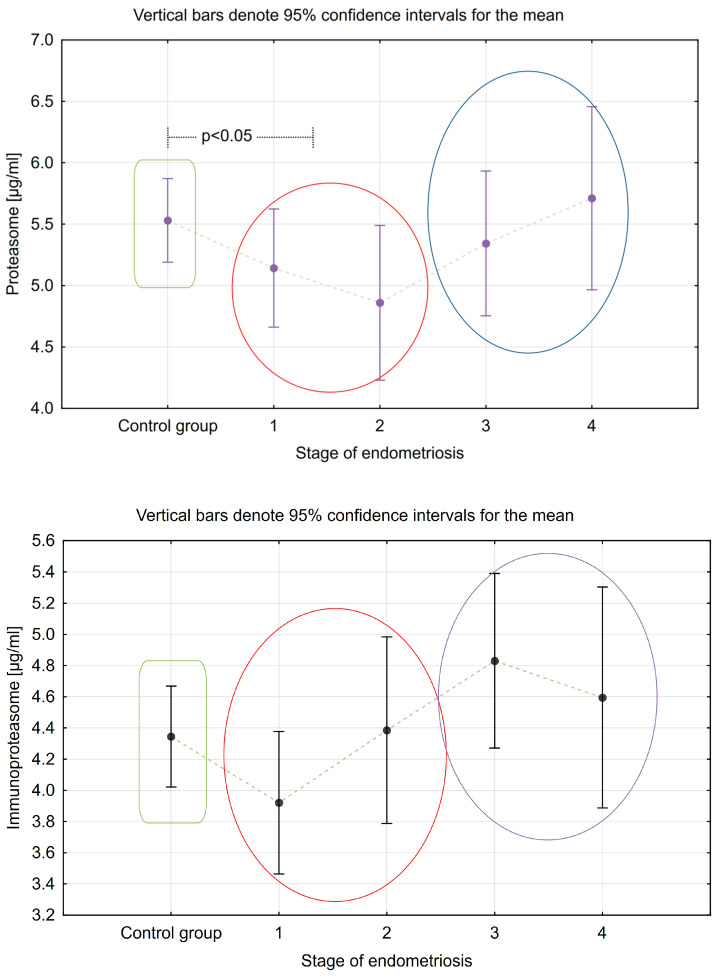
Differences in plasma concentrations of the studied parameters according to the stage of endometriosis.

**Figure 4 ijms-24-14363-f004:**
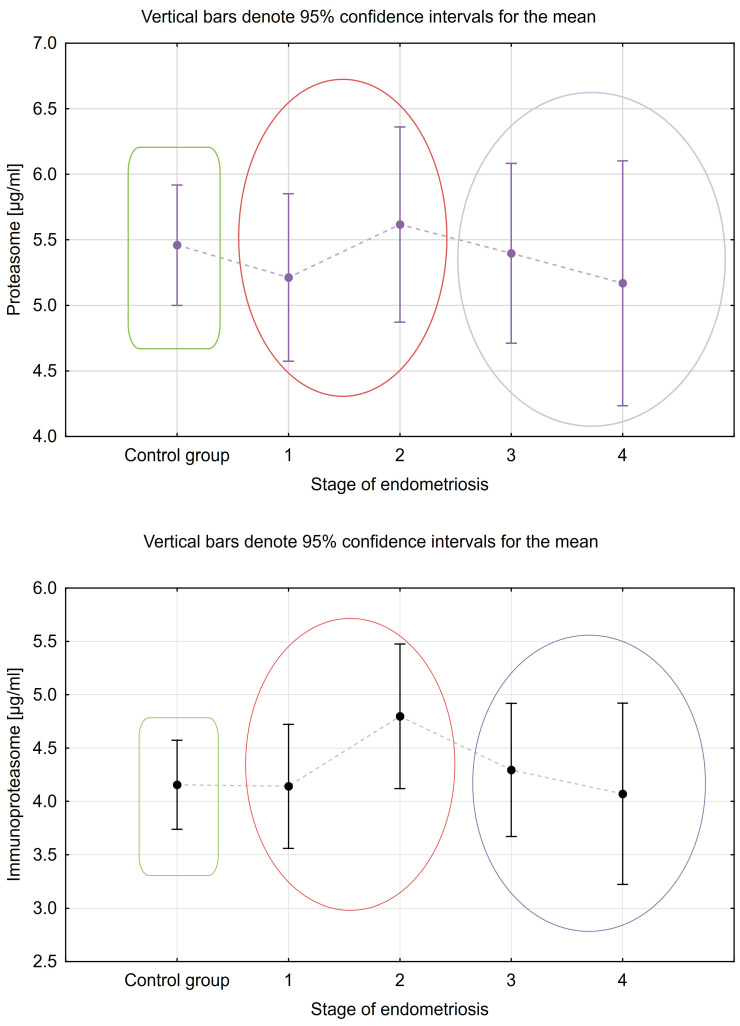
Differences in peritoneal fluid concentrations of the studied parameters according to the stage of endometriosis.

**Figure 5 ijms-24-14363-f005:**
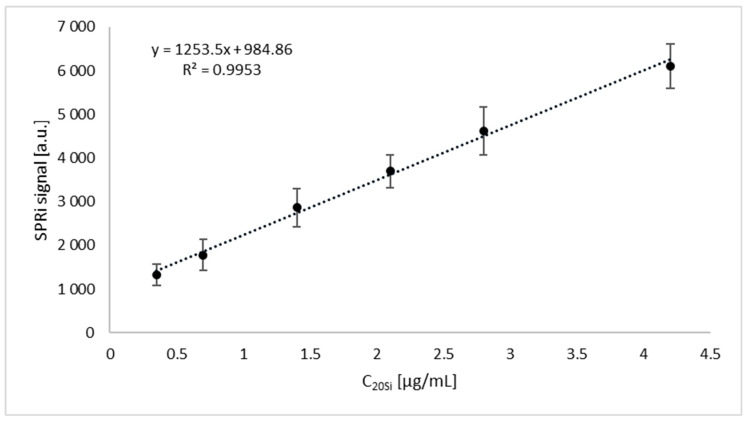
Calibration curve for immunoproteasome 20Si. Concentration of inhibitor = 15 µg/mL, pH = 7.4.

**Figure 6 ijms-24-14363-f006:**
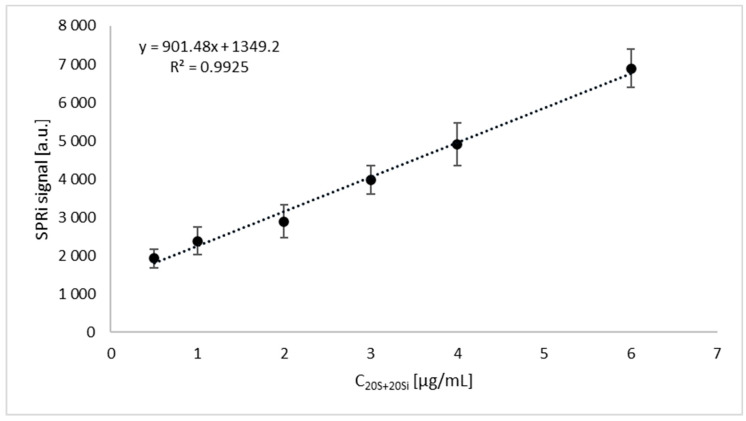
Calibration curve of the sum of the 20S proteasome and the 20Si immunoproteasome. Concentration of inhibitor = 15 µg/mL, pH = 7.4.

**Table 1 ijms-24-14363-t001:** Characteristics of the group of patients in whom plasma was collected.

Variable	Patients with Endometriosis (n = 64)	Patients without Endometriosis (n = 48)	*p*
**Age [years]** Me (IQR)	32.5 (8.0)	32.0 (9.5)	0.301
**Stage I endometriosis**	24 (37.50%)	Not applicable	-
**Stage II endometriosis**	14 (21.53%)	Not applicable	-
**Stage III endometriosis**	16 (25.00%)	Not applicable	-
**Stage IV endometriosis**	10 (15.63%)	Not applicable	-
**Infertility**	36 (56.25%)	24 (50%)	0.672
**First phase of cycle**	46 (71.88%)	37 (77.08%)	0.534
**Second phase of cycle**	18 (28.13%)	11 (22.92%)	
**Ovarian cyst**	31 (48.44%)	0 (0%)	<0.001
**Proteasome concentration** M (±SD)	5.220 (±1.206) µg/mL	5.531 (±1.170) µg/mL	0.174
**Immunoproteasome concentration** Me (IQR)	4.45 (1.48) µg/mL	4.23 (1.34) µg/mL	0.696

Numerical data are presented as Me (median) (IQR—interquartile range) or M (mean) ± SD (standard deviation) depending on the distribution.

**Table 2 ijms-24-14363-t002:** Characteristics of the group of patients in whom peritoneal fluid was collected.

Variable	Patients with Endometriosis (n = 46)	Patients without Endometriosis (n = 29)	*p*
**Age [years]** Me (IQR)	30.5 (7)	31.0 (8)	0.987
**Stage I endometriosis**	15 (32.61%)	Not applicable	-
**Stage II endometriosis**	11 (23.91%)	Not applicable	-
**Stage III endometriosis**	13 (28.26%)	Not applicable	-
**Stage IV endometriosis**	7 (15.22%)	Not applicable	-
**Infertility**	26 (56.52%)	13 (44.83%)	0.527
**First phase of cycle**	27 (58.70%)	22 (75.87%)	0.128
**Second phase of cycle**	19 (41.30%)	7 (24.14%)	
**Ovarian cyst**	27 (58.70%)	0 (0%)	<0.001
**Proteasome concentration** Me (IQR)	5.445 (1.188) µg/mL	5.458 (0.945) µg/mL	0.909
**Immunoproteasome concentration** Me (IQR)	4.583 (1.04) µg/mL	3.936 (1.638) µg/mL	0.284

Numerical data are presented as Me (median) (IQR—interquartile range) or M (mean) ± SD (standard deviation) depending on the distribution of the variables after testing the normality of the distribution using the Shapiro–Wilk test. Categorical data are presented as number (%).

**Table 3 ijms-24-14363-t003:** Distribution of plasma proteasome and immunoproteasome concentrations according to the presence and stage of endometriosis.

Groups	Control Group	Endometriosis Stage			
		I	II	III	IV
Proteasome M (±SD) µg/mL	5.531 ± 1.170	5.143 ± 1.059	4.860 ± 1.632	5.34 ± 1.154	5.711 ± 0.862
Immunoproteasome M (±SD) µg/mL	4.345 ± 0.997	3.921 ± 0.960	4.386 ± 1.731	4.831 ± 1.321	4.596 ± 0.618

Concentrations are presented as M (mean) ± SD (standard deviation).

**Table 4 ijms-24-14363-t004:** Differences in plasma concentrations of the studied parameters according to the presence and stage of endometriosis.

	**Proteasome**	
	**t**	** *p* **
**E1 + E2 vs. E3 + E4** (CONTRAST 1)	−1.68	0.095
**C vs. E1 + E2** (CONTRAST 2)	2.01	**0.047**
**C vs. E3 + E4** (CONTRAST 3)	−0.01	0.990
	**Immunoproteasome**	
	**t**	** *p* **
**E1 + E2 vs. E3 + E4** (CONTRAST 1)	−1.89	0.062
**C vs. E1 + E2** (CONTRAST 2)	0.77	0.445
**C vs. E3 + E4** (CONTRAST 3)	1.31	0.191

One contrast was created for each comparison. E1—Stage I endometriosis, E2—Stage II endometriosis, E3—Stage III endometriosis, E4—Stage IV endometriosis, C—control group.

**Table 5 ijms-24-14363-t005:** Distribution of peritoneal fluid proteasome and immunoproteasome concentrations according to the presence and stage of endometriosis.

Groups	Control Group	I	II	III	IV
Proteasome M (±SD) µg/mL	5.460 ± 1.318	5.213 ± 1.615	5.617 ± 0.433	5.397 ± 1.206	5.169 ± 0.719
Immunoproteasome M (±SD) µg/mL	4.156 ± 1.199	4.142 ± 1.190	4.798 ± 0.609	4.295 ± 1.134	4.072 ± 1.286

**Table 6 ijms-24-14363-t006:** Differences in peritoneal fluid concentrations of the studied parameters according to the presence and stage of endometriosis.

	**Proteasome**	
	**t**	** *p* **
**E1 + E2 vs. E3 + E4** (CONTRAST 1)	0.35	0.729
**K vs. E1 + E2** (CONTRAST 2)	0.13	0.895
**K vs. E3 + E4** (CONTRAST 3)	−0.48	0.635
	**Immunoproteasome**	
	**t**	** *p* **
**E1 + E2 vs. E3 + E4** (CONTRAST 1)	0.825	0.412
**K vs. E1 + E2** (CONTRAST 2)	−1.024	0.309
**K vs. E3 + E4** (CONTRAST 3)	0.083	0.934

One contrast was created for each comparison. E1—Stage I endometriosis, E2—Stage II endometriosis, E3—Stage III endometriosis, E4—Stage IV endometriosis, C—control group.

**Table 7 ijms-24-14363-t007:** Differences in plasma concentrations of the studied parameters according to the selected features.

**PROTEASOME**			
**Feature**	**Present**	**Absent**	** *p* **
**Ovarian cyst** M (±SD) µg/mL	5.458 (±1.027)	5.313 (±1.258)	0.567
**Infertility** M (±SD) µg/mL			
**Patients with endometriosis**	5.005 (±1.256)	5.496 (±1.100)	0.107
**Patient without endometriosis**	5.460 (±1.250)	5.602 (±1.106)	0.265
	**First**	**Second**	** *p* **
**Menstrual cycle phase** M (±SD) µg/mL			
**Patients with endometriosis**	5.249 (±1.209)	5.145 (±1.233)	0.758
**Patient without endometriosis**	5.399 (±1.177)	5.976 (±1.078)	0.153
**IMMUNOPROTEASOME**			
**Feature**	**Present**	**Absent**	** *p* **
**Ovarian cyst** Me (IQR) µg/mL	4.817 (1.291)	4.103 (1.261)	0.017
**Infertility** M (±SD) µg/mL			
**Patients with endometriosis**	4.341 (±1.359)	4.373 (±1.120)	0.912
**Patient without endometriosis**	4.183 (±0.794)	4.507 (±1.159)	0.265
	**First**	**Second**	** *p* **
**Menstrual cycle phase** Me (IQR) µg/mL			
**Patients with endometriosis**	4.364 (1.295)	4.653 (1.826)	0.478
**Patient without endometriosis**	4.174 (1.311)	4.291 (1.336)	0.556

Numerical data are presented as Me (median) (IQR—interquartile range) or M (mean) ± SD (standard deviation) depending on the distribution of the variables after testing the normality of the distribution using the Shapiro–Wilk test.

**Table 8 ijms-24-14363-t008:** Differences in peritoneal fluid concentrations of the studied parameters according to the selected features.

**PROTEASOME**			
**Feature**	**Present**	**Absent**	** *p* **
**Ovarian cyst** Me (IQR)	5.47 (1.054)	5.415 (1.207)	0.987
**Infertility** (M ± SD)			
**Patients with endometriosis**	5.454 ± 1.294	5.227 ± 0.968	0.516
**Patient without endometriosis**	5.693 ± 1.057	5.270 ± 1.504	0.399
**Menstrual cycle phase** Me (IQR)	**First**	**Second**	** *p* **
**Patients with endometriosis**	5.265 (1.714)	5.599 (0.845)	0.284
**Patient without endometriosis**	5.537 (0.984)	5.197 (1.862)	0.665
**IMMUNOPROTEASOME**			
**Feature**	**Present**	**Absent**	** *p* **
**Ovarian cyst** Me (IQR)	4.605 (1.06)	4.048 (1.614)	0.124
**Infertility** (M ± SD)			
**Patients with endometriosis**	4.279 ± 1.072	4.399 ± 1.110	0.714
**Patient without endometriosis**	4.481 ± 1.178	3.891 ± 1.185	0.192
**Menstrual cycle phase** M (±SD)	**First**	**Second**	** *p* **
**Patients with endometriosis**	4.527 (±1.061)	4.053 (±1.069)	0.144
**Patient without endometriosis**	4.239 (±1.211)	3.894 (±1.209)	0.517

Numerical data are presented as Me (median) (IQR—interquartile range) or M (mean) ± SD (standard deviation) depending on the distribution of the variables after testing the normality of the distribution using the Shapiro–Wilk test.

## Data Availability

Data will be available upon contacting corresponding authors.

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
