# Peer review of "Evaluation of Proteasome and Immunoproteasome Levels in Plasma and Peritoneal Fluid in Patients with Endometriosis"

_ijms, 2023, doi:10.3390/ijms241814363_

Round 1
Reviewer 1 Report
Dear Authors,
you have presented very interesting paper, but I have some minor comments about it.
Next to the Correspondence Authors add *
In section affiliation give only postal code (no need to add more information about address)
References in test should be in bracket please place references in one bracket for example line line 75, the references should be in one [6-9] - please check carefully your manuscript
Section materials and methods - line 310 you give information about your new article which described in detail the recruitment of your patients - please first give infomation about even brief about it and at the of the paragraph add this sentence with your recent article [9]
References 9 is as you mentioned your "most recent article" and you should give in section Disscussion information about it that this publication complements your previous study about (...)
Section References - please chcek the Instruction for Authors, how references should be described - for exaple in Journal Articles the Abbreviated Journal Name should be in italic front, next the Year in bold, and Volume in italic front
Author Response
Thank you very much for your valuable comments. I have made the following changes to the manuscript: * signs have been added next to the names of corresponding authors, postcodes have been added next to the relevant addresses, references have been combined in the relevant brackets.
It will obviously be clearer to describe the inclusion and exclusion criteria for the study and then refer to the relevant article. This change has also been implemented in the article.
The publication is part of a series of articles investigating relevant molecules as potential markers of endometriosis. Of course, this is worth mentioning in the discussion- this change was introduced into the manuscript.
Thank you also for pointing out the correctness of the references so carefully. However, I am unable to add the "EndNote template file" suggested by MDPI to my manuscript. Could I ask for help an experienced editorial team?
Reviewer 2 Report
This study looks at levels of 20S proteasome and 20S immunoproteasome in plasma and peritoneal fluid from women with and without endometriosis. Levels of proteasome and immunoproteasome were detected using Surface Plasmon Resonance imaging. While this study found no difference in 20S proteasome and 20S immunoproteasome in women with or without endometriosis, it is postulated that plasma proteasome levels may be indicative of progression of endometriosis.
I am uncertain if this is a complete manuscript or if inclusion of results from cycle phase and infertility status (referred to in the discussion – lines 274-277) as well as the future planned work (last paragraph of the discussion) would improve the strength of this manuscript and provide a stronger research argument.
The abstract refers to ‘patients with suspected endometriosis’, what does this mean? Were these patients symptomatic. Were the control group also women with symptoms consistent with endometriosis but with no evidence of disease – I believe this is important to establish as if this is the case then the ‘controls’ could have other underlying pathologies.
Reference are required to be added in a number of locations throughout the manuscript –
Introduction line 76 –‘laparoscopy is no longer the diagnostic gold standard…’
Discussion line 274 – reference to previous work
In the discussion line 245 ‘…proven effective for these both diseases’ correct the grammar
A methodological question – was plasma and peritoneal fluid levels compared within the same patient, if so what was the association?
Author Response
Thank you very much for your valuable comment. The manuscript is complete at this point, we have analyzed the differences in proteasome and immunoproteasome concentrations on the material we dispose of. Cycle phase and infertility status were also taken into account, but we did not find differences in the concentrations of the parameters studied in this area. We are in the process of planning a study based on a larger group, the results of which will certainly complement our study.
The phrase 'patients with suspected endometriosis' refers to patients who were referred for elective laparoscopy due to infertility, pelvic pain or ovarian cysts. This cohort was divided into groups who were diagnosed with endometriosis and those in whom the disease was not confirmed. The control group was therefore patients who also developed symptoms. A detailed description of the inclusion criteria for the study can be found in lines 314-320.
In line 79, references to the latest ESHRE guidelines have been added. On the other hand, line 274 was intentionally left without references - this is the possible mechanism behind the difference in plasma immunoproteasome concentration obtained.
In line 245, a grammatical error has been corrected- thank you for your comment.
Due to the fact that the group from which peritoneal fluid and plasma were collected was not exactly the same, no comparisons were made between the concentrations of the studied parameters in the same patients. However, this is a valuable observation that we will take into account in further research.
Reviewer 3 Report
The involvement of proteasome in endometriosis pathology has been previously suggested, but this is the first attempt to measure proteasome levels from plasma of patients with and without endometriosis.
Unfortunately, the study results are largely negative as the only statistically significant difference between proteasome levels was between stage I and II endometriosis compared to the control group, and even this p-value is very close to 0.05. However, in the reviewer's opinion, it is very important to publish negative results as well, as this helps to exclude the investigated compounds from further studies.
However, some issues would need clarification:
Whether it is necessary to add four decimal places to the p-values.
Tables: please define the abbreviations used in the tables (Me, M, IQR).
Tables 1 and 2: what does mean “first/second phase of cycle”? Does that mean follicular/luteal phase? If samples were collected throughout the cycle, it does not make sense to report the median day of the menstrual cycle.
Is it necessary to describe SPRI in such detail in this manuscript (e.g. lines 362-386, 396-403)? I would recommend referring to published materials (e.g. reference 25, Gorodkiewicz et al.) whenever possible and describing here only what is specific to this study.
The presence of endometriomas usually indicates a more severe stage of endometriosis. In the current study, 31 patients had ovarian cysts (Table 1), but 26 patients had severe endometriosis (III-IV stage). This discrepancy needs clarification.
Would it be possible to combine tables showing similar data on plasma and peritoneal fluid (i.e. Tables 1 and 2, 3 and 5 etc.)?
Author Response
Thank you very much for your valuable comment.
Of course, it will be sufficient if the p-value is calculated to 3 decimal places - the correction has been made in the paper.
The abbreviations under the tables have been clarified.
Obviously, the terms first and second phase of the cycle refer to the follicular and luteal phases. We also compared the day of the cycle to confirm the appropriate selection of groups, but we agree that this information is not necessary, so it has been removed.
The SPRI method has been described in detail, as not all details are identical to previous work published by our team. However, the construction of the biosensor remains unchanged, so here we have made a change and included a link to the relevant article.
Thank you for pointing out the differences in the number of patients with endometrial cysts and those with advanced endometriosis. In fact, there is an inaccuracy here. Not all cysts that were detected turned out to be endometrial cysts - for clarity, the term has been replaced by 'ovarian cysts in patients with endometriosis'. A detailed explanation has also been added to the manuscript. However, this does not change the validity of the conclusions reached.
In this study, we wanted to clearly distinguish the differences in the concentrations of the studied parameters in plasma and peritoneal fluid, hence, in our opinion, the presentation of the obtained results in separate tables is more understandable. We propose to leave this form, but of course, if in the reviewer's opinion this change is necessary, it will be implemented.